# Improving quality of surgical and anaesthesia care at hospital level in sub-Saharan Africa: a systematic review protocol of health system strengthening interventions

Nataliya Brima [1], Justine Davies,[2] Andrew JM Leather[1]

¹King's Centre for Global Health & Health Partnerships, King's College London, London, UK
²Institute of Applied Health Research, University of Birmingham, Birmingham, UK

**Correspondence to**
Mrs Nataliya Brima;
nataliya.brima@kcl.ac.uk

## ABSTRACT

**Introduction** Over 5 billion people in the world do not have access to safe, affordable surgical and anaesthesia care when needed. In order to improve health outcomes in patients with surgical conditions, both access to care and the quality of care need to be improved. A recent commission on high-quality health systems highlighted that poor-quality care is now a bigger barrier than non-utilisation of the health system for reducing mortality.

**Aim** To carry out a systematic review to provide an evidence-based summary of hospital-based interventions associated with improved quality of surgical and anaesthesia care in sub-Saharan African countries (SSACs).

**Methods and analysis** Three search strings (1) surgery and anaesthesia, (2) quality improvement hospital-based interventions and (3) SSACs will be combined. The following databases EMBASE, Global Health, MEDLINE, CINAHL, Web of Science and Scopus will be searched. Further relevant studies will be identified from national and international health organisations and publications and reference lists of all selected full-text articles. The review will include all type of original articles in English published between 2008 and 2019. Article screening, data extraction and assessment of methodological quality will be done by two reviewers independently and any disputes will be resolved by a third reviewer or team consensus. Three types of outcomes will be collected including clinical, process and implementation outcomes. The primary outcome will be mortality. Secondary outcomes will include other clinical outcomes (major and minor complications), as well as process and implementation outcomes. Descriptive statistics and outcomes will be summarised and discussed. For the primary outcome, the methodological rigour will be assessed.

**Ethics and dissemination** The results will be published in a peer reviewed open access journal and presented at national and international conferences. As this is a review of secondary data no formal ethical approval is required.

**PROSPERO registration number** CRD42019125570.

## INTRODUCTION

The urgent need for surgical and anaesthesia care continues to be neglected, with over 5

### Strengths and limitations of this study

- ► A comprehensive search strategy for original articles and grey literature both published and unpublished was created and included.
- ► The inclusion of all fields of surgery in the search strategy will ensure that no health system strengthening interventions implemented within sub-Saharan Africa (SSA) region is missed.
- ► It is possible that the included studies (and reported data) will be too few to address the objectives of this study.
- ► There is the potential that the data retrieved from different studies may not be applicable to every SSA country context due to differences in the healthcare system and therefore will need to be carefully adapted to the context of another country.
- ► Although every effort will be made to address the publication bias, we acknowledge that unsuccessful quality improvement efforts and unwanted side effects are rarely published, and as a result this review may not reflect the real current experience of failed attempts in the field of quality improvement of surgical care.

billion people worldwide being deprived access to safe, affordable and timely surgical and anaesthesia care when needed.[1] Sub-Saharan African countries (SSACs) are disproportionally affected where 93% of the population do not have access to basic surgical care.[1] Mortality and morbidity from treatable surgical diseases are high and continues to grow. In 2010, an estimated 32.9% of all lives lost were due to surgical burden of disease.[2] Untreated surgical diseases are among the top 15 causes of physical disability worldwide, with injuries accounting for 15% of all ill-health worldwide and 90% of deaths from injuries occurring in low-income or middle-income countries (LMICs),[3] 85% of all SSACs are classified as either low-income or middle-income

BMJ

countries. The poorest countries account for 34·8% of the global population, yet annually, only 6% of surgical procedures worldwide take place in these countries[4] and sub-Saharan Africa (SSA) region has the greatest unmet need for surgical procedures.[1] Surgical patients in Africa are two times as likely to die after surgeries, despite being younger, lower-risk profile and developing fewer complications, compared with the global average.[5]

Annually, 1.8 million deaths in LMICs could be surgically averted, with 1.4 million treatable by basic surgical care delivered at first-level hospitals and 0.4 million treatable by advanced surgical care delivered in specialised facilities.[6] This will require strong health systems with effective referral from community to primary care and onwards to secondary and tertiary hospital care[1] in order to address the barriers in accessing quality care. Recently, the Lancet Global Health Commission on High-Quality Health Systems in Sustainable Development Goals Era highlighted the low quality and performance of many LMIC health systems and placed emphasis on the prioritisation of high-quality care to reduce avoidable mortality and morbidity.[7]

In this systematic review we will identify, critically appraise and synthesise the existing published evidence on health system interventions that have focused on quality surgical and anaesthesia care in hospitals in SSACs. This study will inform future efforts to improve the quality of hospital-based care for patients requiring surgical and anaesthesia care.

## METHODS AND ANALYSIS

The systematic review will be conducted, and the data reported in accordance with the Preferred Reporting Items for Systematic Review and Meta-Analysis (PRISMA) statement.[8] This protocol was developed following the PRISMA Protocol 2015 checklist,[9 10] (online supplementary file 1). Amendments to the protocol are not anticipated but will be reported in the publication of the results, should they occur.

## AIM

To provide an evidence-based summary of hospital-based interventions associated with improved quality of surgical and anaesthesia care in SSACs.

## Objectives

- ► To systematically identify studies that evaluate interventions aimed at improving quality of surgical and anaesthesia care at hospital level in SSACs.
- ► To capture, categorise and assess the study characteristics (setting and intervention).
- ► To capture and categorise all outcomes reported under the following categories: clinical, process, implementation or other.
- ► To assess methodological rigour of studies with mortality as a primary outcome (which is defined as mortality within 30 days defined for this review).

- ► To record whether process and implementation outcomes were assessed.

## Search strategy

The search string strategy was developed by the main author and checked by a research librarian for errors and omissions. During the development process, the search was tested and retested to ensure optimal sensitivity and specificity and search terms refined accordingly. The search strategy was developed in MEDLINE (online supplementary appendix 1). The search terms used intended to cover a wide range of studies that evaluated interventions aimed at improving the quality of surgical and anaesthesia care at hospital level in SSACs. The three distinct search strings that were combined to identify these studies are (1) surgical and anaesthesia care, (2) quality improvement hospital-based interventions and (3) SSACs.

Surgical and anaesthesia care will include all branches of surgery (cardiothoracic, ear, nose and throat (ENT), general, neonatal, obstetric, ophthalmic, orthopaedic, paediatric and urology), including operative and non-operative aspects of care, on the presumption that any branch of surgery may have relevant transferable interventions within a low resource setting. For the purpose of this review only studies that explicitly refer to a surgical procedure or a surgical patient will be included. Trauma (both explicitly focusing and included as part of the intervention) will be excluded, as it will be covered in a separate systematic review.

For the purposes of this review health system strengthening intervention (HSSI) is defined as any actions or strategies taken to improve the quality of healthcare delivery or patient outcomes. This will include studies that either strengthen the whole surgical system within a hospital or contribute to one aspect of surgical and anaesthesia care. The WHO Health System Framework will be used to categorise HSSIs.[11]

Hospital-based interventions are defined as any intervention that is delivered within a tertiary/teaching/national or a secondary/district/regional care hospital environment, aimed at improving quality of care, that directly or indirectly involves care delivery to patients or by staff. Both hospital-wide (hospital documentation, early warning scores, intensive care outreach) and individual departmental (emergency department, surgical wards, theatre, high dependency unit (HDU), intensive treatment unit (ITU), outpatient, radiology) interventions will be included.

The term SSACs will include all countries listed as SSA counties by the World Bank for the 2019 fiscal year.[12]

To combine terms within each search string, the Boolean operator 'OR' was used. To incorporate all variations of the same term, truncation and wildcards were employed. Medical Subject Headings terms and free text were also used. To focus the search on these required terms, phrase and adjacency searching, using proximity operator filters and limits, was exploited. Any additional

key terms identified during the literature search were added to the appropriate string to ensure it was maximally inclusive. The three strings were then combined using the Boolean operator 'AND'.

This search strategy will be adapted to EMBASE, Global Health, CINAHL and Web of Science databases. A reference list of all included articles will be hand-searched for further relevant studies.

To identify unpublished studies, the Scopus literature databases will be searched. In addition, first 50 hits from Google search, documentation and reports of relevant national and international health organisations such as Comic Relief, DFID, Grand Challenges Canada, the Bill and Melinda Gates Foundation, HIVOS (*Humanistisch Instituut voor Ontwikkelingssamenwerking*), World Vision, the Robert Wood Johnson foundation, United Nations Children's Fund (UNICEF) and GOAL will be searched.

The choice of information sources for this search were made based on the recommendation from the current literature, to ensure the optimal combination of databases and other search resources available needed to guaranty adequate and efficient coverage.[13]

The search process will be recorded using a Database literature search log (online supplementary appendix 2).

### Eligibility criteria

All eligible articles published in English will be included, from 2008 to 2019. The inclusion and exclusion criteria are illustrated in table 1 below.

### Data extraction

All potential articles will be initially collated in Endnote X8 for de-duplication of results. All remaining studies will be loaded into Rayyan online opensource web application[14] and initially screened by title and abstract by a team of two reviewers independently, according to the inclusion and exclusion criteria (table 1) and by using the Systematic Review Abstract and Full-Text Screening Criteria form (online supplementary appendix 3). All reviewers were appropriately trained in systematic review methods. If either of the reviewers include an abstract, it will be promoted to consideration for full-text review. The full-text review will be done by the first author (NB) and another reviewer independently, using inclusion and

| Table 1 | Inclusion and exclusion criteria | |
|---|---|---|
| | **Include** | **Exclude** |
| Type of article | All peer-reviewed research articles Non-research reports from national or international health organisations, dissertations/theses, books/book chapters, conference abstracts and research in progress from the grey literature | Unstructured reviews or overviews, theoretical papers, commentaries or opinion papers, Case studies, audits, editorials/letters/comments, newspapers/trade journals, literature reviews Guidelines, strategies and policies from national or international health organisations |
| Type of conditions | Any surgical and anaesthesia care (operative or non-operative); type of presentation (elective or emergency); sub-speciality surgical or anaesthesia care (including perioperative medicine and pain management) | Trauma/injury care Studies on cosmetic and aesthetic surgical care and sports medicine |
| Type of population | General population Population with specific surgical diseases or conditions Adults, neonatal and children's surgery | Animals |
| Care setting | Hospital setting and SSACs | Studies that are not conducted in hospital-based settings Studies that took place outside of SSA |
| Type of design | Interventional studies* | Observational studies |
| Subject of study | Quality improvement of surgical care through the following areas: ► Service delivery ► Health Workforce ► Information ► Financing ► Leadership/governance | ► Studies that did not assess outcomes ► Medical device production and new clinical technological devices ► Introduction of new procedures ► Medical products, vaccines and technologies |

*Interventional studies are often prospective and are specifically tailored to evaluate direct impacts of treatment or preventive measures on disease.[17]

SSA, sub-Saharan Africa; SSACs, sub-Saharan African countries.

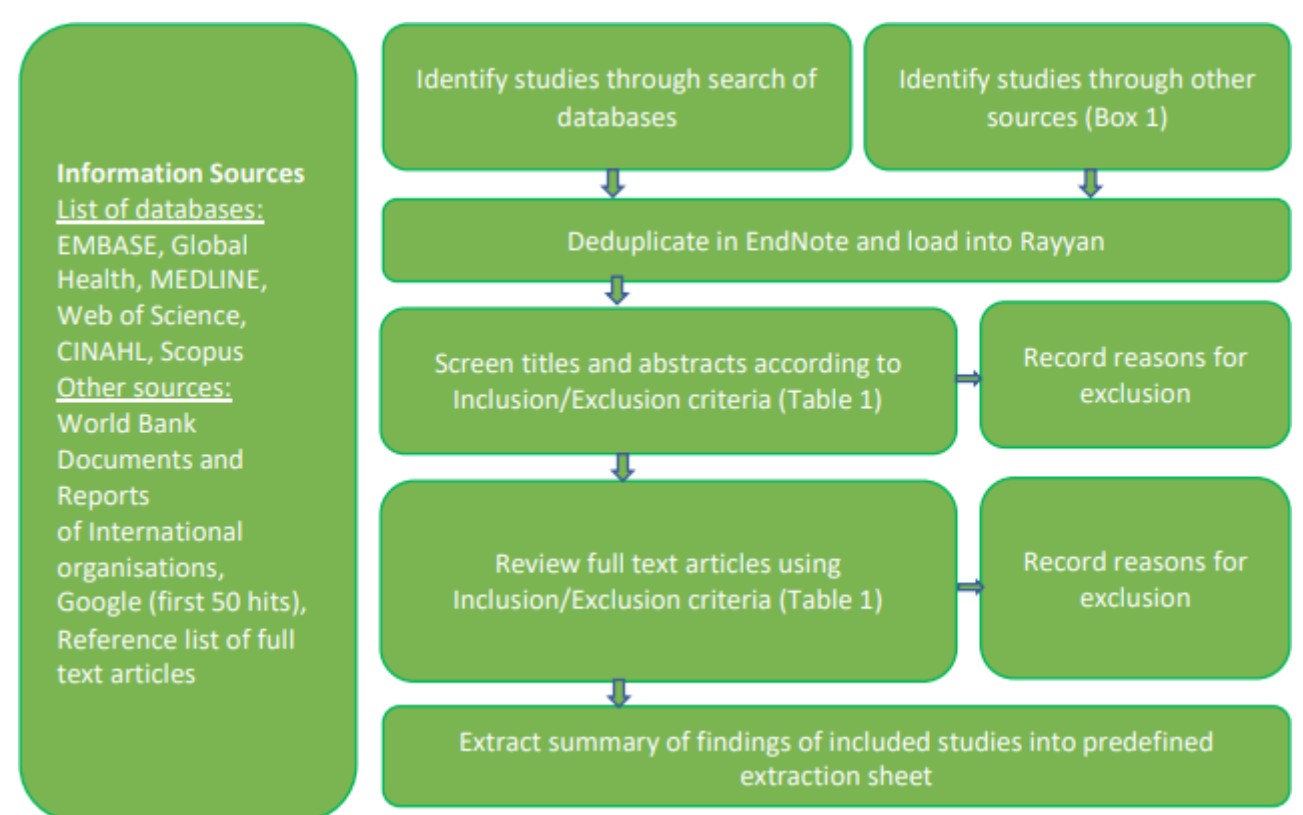

**Figure 1** Data search and extraction flow diagram.

exclusion criteria. Data elements will be extracted into a predefined data extraction form (online supplementary appendix 4), which will be piloted before finalising the elements to be included. This includes information on authors, publication dates, country of intervention, healthcare setting, type of intervention, methodology or design, patient/intervention size, clinical/process/implementation/other outcomes and the impact of the intervention. The data extracted by two reviews will be compared and if any discrepancies arise, consensus will be sought among the study team (figure 1).

### Outcomes

Three type of outcomes, clinical, process and implementation, will be collected. The primary outcome for this review will be mortality (in-hospital perioperative mortality and post intervention mortality within 30 days). All other outcomes will be secondary: clinical outcome such as major and minor complications as defined by the Clavien-Dindo Classification (table 2) and any other clinical outcomes identified during the course of reviewing the manuscripts. Process outcomes will include all outcomes relevant to interventional studies included in this review (waiting times, length of hospital stay, blood availability, and so on). Implementation outcomes will be determined using a 'working taxonomy' of eight conceptually distinct implementation outcomes produced by Proctor et al[15] (table 3). If any other outcomes are reported that cannot fit into the three categories above, they will be recorded

| Table 2 | Clavien-Dindo Classification of complications |
|---------|-----------------------------------------------|
| **Grade** | **Definition** |
| I | Any deviation from the normal postoperative course without the need for pharmacological treatment or surgical, endoscopic and radiological interventions. Allowed therapeutic regimes are as follows: drugs as anti-emetics, anti-pyretics, analgesics, diuretics, electrolytes and physiotherapy. This grade also includes wound infections opened at the bedside. |
| II | Requiring pharmacological treatment with drugs other than such allowed for grade I complications. Blood transfusions and total parenteral nutrition are also included. |
| III | Requiring surgical, endoscopic or radiological intervention. |
| IIIa | Intervention not under general anaesthesia. |
| IIIb | Intervention under general anaesthesia. |
| IV | Life-threatening complication requiring ICU management. |
| IVa | Single-organ dysfunction (including dialysis). |
| IVb | Multiorgan dysfunction. |
| V | Death of a patient. |

**Table 3** Implementation outcomes

| | Implementation outcome (Proctor et al)[15] | Definition |
|---|---|---|
| 1 | Acceptability | Degree to which the intervention is perceived agreeable, acceptable or satisfactory |
| 2 | Adoption | Intention and willingness to apply the intervention |
| 3 | Appropriateness | Perceived relevance of the intervention to be relevant to the problem |
| 4 | Feasibility | Extent to which the intervention can be successfully applied |
| 5 | Fidelity | Extent to which the intervention is completed as originally intended |
| 6 | Implementation Cost | Intervention time and cost |
| 7 | Penetration | Spread into practice, for example, proportion of eligible patients who received the intervention |
| 8 | Sustainability | Extent to which a new intervention is routinely used in practice |

under the heading 'other outcomes', described, and categorised, if possible, after discussion among authors.

## Risk of bias

Risk of bias of individual studies and publication bias will be addressed by designing a thorough search strategy to find relevant studies. The search strategy will be assessed by checking the results against known articles which are relevant. To address publication bias the grey literature will be searched, as described above, to include all relevant studies which are not reported in peer-reviewed journals. The following type of publications will be included: dissertations/theses, books/book chapters, conference abstracts and research in progress. Also, all potential full-text articles will be screened by two reviewers. For studies that reported their main outcome as mortality the reviewers will independently assess the risk of bias as part of the Grading of Recommendations Assessment, Development, and Evaluation (GRADE) system[16] of rating quality of evidence and grading strength of recommendations in systematic reviews.

Disagreements between the reviews over the risk of bias level (serious or very serious) will be resolved by discussion with involvement of a third reviewer where necessary. Due to expected high heterogenicity of the data extracted, a meta-analysis will not be feasible.

## Data synthesis

For all outcomes all items that will be collected using predefined collection form (online supplementary appendix 4) will be summarised and the methodological approach used in the studies will be identified. Descriptive statistics and narrative synthesis of data will be undertaken to appreciate the diversity of research approaches that are used for assessment of quality of surgical and anaesthesia care within hospital. Those studies that were collected under the heading 'other', we will look for common themes and then compare these themes to our three defined categories of outcomes.

In addition, the studies reporting mortality as the main outcome of the interventions will be assessed using GRADE criteria for rating quality of evidence and grading

strength of recommendations in systematic reviews.[16] The criteria are as follows: study design, risk of bias, imprecision, inconsistency, indirectness, and magnitude of effect. They will be summarised in summary of findings table (online supplementary appendix 5). Based on this information quality of evidence will be rated and detailed information about the reason for the quality of evidence rating will be recorded. Outcomes will be categorised as strong or weak according to the quality of the supporting evidence.

## Patient and Public Involvement

No patient involved.

## Ethics and dissemination

This study is a part of a wider National Institute of Health Research (NIHR) Global Health Research Unit on Health System Strengthening in SSA (ASSET), King's College London (GHRU 16/136/54) project. The results will be communicated and disseminated during meetings with key stakeholders at national, district and community levels, that will be on-going throughout the project. The results will inform the development and implementation of a quality improvement intervention to improve quality of surgical and anaesthesia care in SSACs in general and in Sierra Leone in particular. The results of the review will be published in a peer reviewed journal and presented at national and international conferences. As this is a review of secondary data no formal ethical approval is required.

To our knowledge this is the first systematic literature review to synthesise the information on HSSIs to improve the quality of hospital-based surgical and anaesthesia care in SSACs. The findings from this systematic review will provide evidence to inform the development, design and implementation of the HSSIs to improve the quality of health care for surgical patients at hospital level in SSACs. It will also provide valuable information for policy makers, healthcare managers and clinicians.

**Acknowledgements** This study is part of NIHR Global Health Research Unit on Health System Strengthening in sub-Saharan Africa (ASSET),King's College London

(GHRU 16/136/54) and supported by the King's Centre for Global Health and Health Partnership. The authors thank the library services at King's College London, UK.

**Contributors** NB, AJML and JD conceived the idea for the study and designed the protocol. NB drafted the protocol and is the guarantor of the review. All authors provided feedback on the study design and review methods and contributed to the final manuscript.

**Funding** This systematic review is supported by the NIHR Global Health Research Unit on Health Systems Strengthening in Sub-Saharan Africa, King's College London (GHRU 16/136/54). The funding body had no input into the study design.

**Competing interests** None declared.

**Patient and public involvement** Patients and/or the public were not involved in the design, or conduct, or reporting, or dissemination plans of this research.

**Patient consent for publication** Not required.

**Provenance and peer review** Not commissioned; externally peer reviewed.

**ORCID iD**
Nataliya Brima http://orcid.org/0000-0002-6930-5166

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
