## [Reviewer comments · BMJ Open]

ARTICLE DETAILS

TITLE (PROVISIONAL)	Improving quality of surgical and anaesthesia care at hospital level in Sub-Saharan Africa: a systematic review protocol of health system strengthening interventions.
AUTHORS	Brima, Nataliya; Davies, Justine; Leather, Andy

VERSION 1 – REVIEW

REVIEWER	Henry Rice Duke University
REVIEW RETURNED	15-Jan-2020

GENERAL COMMENTS	This protocol by Brima et al. is an extremely well written summary of a planned systematic review of interventions to improve surgery and anesthesia care in Sub-Sahara African countries. The co-authors are all well-recognized experts in these areas. Overall there are no substantial concerns that would limit publication of this protocol. My only question for the authors, and this is simply one of style, is whether the implementation framework of Proctor is the best for this analysis. Much of the surgical and anesthesia literature does use this framework, although alternatively the authors could consider using the Consolidated Framework for Implementation Research. This may be more cumbersome and less helpful for this particular project, but the CFIR is gaining increasing prominence as the preferred framework for implementation studies.
--

REVIEWER	Dr Tom Bashford NIHR Global Health Research Group on Neurotrauma, University of Cambridge, UK I am a Visiting Lecturer for the MSc in Global Surgery at King's College London, which is run by the senior author of this paper and for which I receive an honorarium. I have no other competing interests to declare.
REVIEW RETURNED	22-Jan-2020

GENERAL COMMENTS	Thanks for the opportunity to review this timely effort to synthesise the growing literature on health systems strengthening as it pertains to surgical care in sub-Saharan Africa. The protocol represents a robust attempt to draw together the existing evidence using an established and appropriate methodology. I have four fundamental comments, and several minor points of clarification which are listed specifically below. My four fundamental questions are around the outcomes, the approach to data synthesis, the bias, and the breadth.
--

	1. I was left slightly confused by the outcomes of interest, and the way in which the term 'outcomes' was used throughout the text. The abstract states clearly that the primary outcome is mortality, and that methodological rigour will be assessed only for the primary outcome (page 4 line 32). Within the search strategy a health-systems strengthening intervention is then defined as an action taken to improve the quality of health care delivery AND patient outcomes (page 7 line 5). On the same page, (line 17) tis is then apparently extended to include staff outcomes. Later, in the 'Outcomes' section (page 10 line 41) three types of outcomes are listed: clinical, process and implementation. Finally, the main clinical outcomes are planned to be assessed using GRADE criteria (page 12 line 22). Having recently completed a systematic review on quality improvement interventions, I am very aware how heterogeneous the data can be - as the authors themselves note (page 12 line 7). I can understand the clarity that a focus on a primary outcome of mortality would bring: it would greatly reduce the number of articles but also allow a more direct comparison of the effects of the proposed interventions. Widening the concept of outcomes to include a range of clinical, process, and implementation metrics will allow a much richer and more complete assessment of the literature but make comparison much more challenging. There is also a question about the balance between an inductive as opposed to a hypothetico-deductive approach. Common 'outcomes' may emerge from the literature which challenge those postulated by the authors. It should be possible to screen out papers which make no mention of any outcome assessment (as proposed page 9 line 4) and then compare the themes which emerge from the literature to those used a priori. I think some mention also needs to be made that different metrics will need to be manipulated to allow direct comparison (eg if survival vs morality is reported, then the sign needs inverting to allow comparison - if data are to be converted into a common format such as 30-day mortality then this should be explained, or if all studies will be excluded apart from those which explicitly use 30 day mortality then this also needs to be stated). I am certain the authors are very clear on this, but the terminology used throughout the paper could be tightened to make it more explicit to the reader. Is the primary outcome of mortality a defining characteristic of the search, or a benchmark for comparing the (probably small) subset of papers who report it? If the latter, will this analysis of small subset be the most useful lesson to be gained from this literature? 2. Developing this theme, the proposed 'narrative synthesis' is defensible and reasonably standard as is the use of only descriptive statistics. However an alternative might be something like a 'best-fit framework' analysis which has been used to synthesise qualitative literature (https://bmcmmedicine.biomedcentral.com/articles/10.1186/1741-7015-9-39). This might allow the authors to assess the extent to which the literature they identify supports their proposed models of clinical, process, or implementation outcomes and, if it does not, critically appraise where their proposed models are challenged. 'Quality' is a difficult term to define and may have components which pull in different directions: it is easy to conceive an intervention that reduces cost, improves access, but worsens individual clinical outcome despite improving population health outcomes. How would the 'quality' of such an intervention be addressed by the current study strategy?
--	---

3. This finally leads to the question of bias. The end goal of this research is to ultimately inform the design and delivery of a future intervention (page 13 line 4). As such, learning from failure may be as crucial as learning from success, but there is a huge inherent bias in published accounts of quality improvement efforts whereby unsuccessful attempts, or unwanted side-effects, are rarely published. The greatest bias, to my mind, of this data will be that it will largely synthesise the current experience of success but speak very little to the current experience of failure. I think the authors have taken every care to address bias in their study design, but would prefer to see a greater acknowledgement of these fundamental issues in their discussion of general limitations and also specific issues with bias.

4. The study quite rightly aims to assess both surgical and anaesthetic care, and is also framed very much within the (completely appropriate, in my view) paradigm of health systems strengthening. A key process in systems thinking is the definition of the system of interest and its boundary, and this is not dwelt on in terms of how a health systems strengthening intervention (HSSI) might affect perioperative care. Page 7, line 17-20 give a good idea of the boundaries considered as the 'surgical system' which include anaesthesia, intensive care, and the emergency department. These are appropriately reflected by the search terms in Appendix 1. Once these papers are identified, however, how will the research team navigate this system boundary? Would - for example - an intervention purely aimed at improving ITU care in a hospital which provides surgical care be included, even if no explicit mention of surgery is made? This seems to me to be the hardest part of synthesising this data. I think the answer will be heuristic and difficult to define explicitly before the literature is reviewed. However I think a bit more explicit thought to show how these systems boundaries are intended to be managed would be useful, and the link between deductive vs inductive reasoning developed in line with Comment 1 above. Completely defining all boundaries and terms at the start obviously maximises the ability of the protocol to be repeated, but does not allow for iterative refinement based on exposure to the actual data.

Specific points:

Page 4 line 10: is this the Lancet Commission on health systems? I think a bit more specificity would be useful (eg 'a recent Lancet Commission on high quality health systems').

Page 4 line 49: some of the points raised above need to be somehow addressed in the limitations.

Page 5 line 24: classifies = classified?

Page 5 line 30: less = fewer?

Page 5 line 38-42: although familiar, this is just one model to improve access to surgical care.

Page 6 line 19: these objectives could be revised for clarity in light of major moments above. If you are only assessing the methodological rigour of studies with primary clinical outcomes does this mean you are excluding all studies for which their primary outcome was not clinical? Or just not assessing their rigour? This seems a slightly narrow focus and will limit the ultimate utility of the results.

	Page 6 line 56: it might be worth quickly explaining how you will exclude trauma as it seems likely that some 'systems strengthening' efforts may act on multiple surgical services. If an improvement is explicitly focussed on trauma then I am assuming ti all be excluded, but what will you do with those which focus on multiple surgical subtypes (such as WHO checklist implementation)? These may include trauma data, so will they be excluded, or will you try to unpick the trauma component of the data, or will you include them as being predominantly 'surgery' focussed as opposed to trauma? Page 7 line 5: Is this the defining criteria - any study that aims to improve quality AND patient outcomes? To me this is quite different from systems strengthening efforts which might look at staff, family, or institutional outcomes. This is part of the wider question about clarity of outcomes mentioned above. Page 7 line 17: mention of staff here seems in contradiction to line 5 above? Page 8 line 57: Are interventional studies the only way to asses system strengthening efforts? This has a very positivist, 'cause-and-effect' construction which is well suited to more common discrete medical interventions (such as pharmaceutical testing). Is this necessary if looking to assess quality improvement, system strengthening etc?
--	---

VERSION 1 – AUTHOR RESPONSE

Reviewer Comments	Our response
Reviewer 1: This protocol by Brima et al. is an extremely well written summary of a planned systematic review of interventions to improve surgery and anaesthesia care in Sub-Saharan African countries. The co-authors are all well-recognized experts in these areas. Overall there are no substantial concerns that would limit publication of this protocol. My only question for the authors, and this is simply one of style, is whether the implementation framework of Proctor is the best for this analysis. Much of the surgical and anaesthesia literature does use this framework, although alternatively the authors could consider using the Consolidated Framework for Implementation Research. This may be more cumbersome and less helpful for this particular project, but the CFIR is gaining increasing prominence as the preferred framework for implementation studies.	Thank you for this positive statement about our SR protocol Thank you, this is a valuable point. Proctor and colleagues established a core set of implementation outcomes which have been widely used since their paper was published. We selected this framework as we were keen to analyse if reported hospital-based quality improvement interventions had been implemented rigorously using not just clinical and process

	outcomes – but importantly, also implementation outcomes. The Consolidated Framework for Implementation Research (CFIR), that is composed of 39 constructs over five major domains (interventional characteristics, inner settings, outer setting, characteristics of individuals and implementation process) was developed to guide systematic assessment of multilevel implementation contexts to identify factors that might influence intervention implementation and effectiveness. Initially we had considered to include a wider assessment of the implementation process and we have given this further careful consideration as a team. However, we have decided to prioritise the assessment of implementation outcomes in this systematic review and use Proctor framework, given its previous wide scale use, focus on implementation outcomes, and the fact that other researchers with whom we are working are utilising this approach in their research – allowing comparison of constructs across projects.
Reviewer: 2	
Thanks for the opportunity to review this timely effort to synthesise the growing literature on health systems strengthening as it pertains to surgical care in sub-Saharan Africa. The protocol represents a robust attempt to draw together the existing evidence using an established and appropriate methodology. I have four fundamental comments, and several minor points of clarification which are listed specifically below. My four fundamental questions are around the outcomes, the approach to data synthesis, the bias, and the breadth. 1. I was left slightly confused by the outcomes of interest, and the way in which the term 'outcomes' was used throughout the text. The abstract states clearly that the primary outcome is mortality, and that methodological rigour will be assessed only for the primary outcome (page 4 line 32).	Thank you for this affirmation of our review.

Within the search strategy a health-systems strengthening intervention is then defined as an action taken to improve the quality of health care delivery AND patient outcomes (page 7 line 5).

On the same page, (line 17) tis is then apparently extended to include staff outcomes.

Later, in the 'Outcomes' section (page 10 line 41) three types of outcomes are listed: clinical, process and implementation.

Finally, the main clinical outcomes are planned to be assessed using GRADE criteria (page 12 line 22).

Having recently completed a systematic review on quality improvement interventions, I am very aware how heterogeneous the data can be - as the authors themselves note (page 12 line 7). I can understand the clarity that a focus on a primary outcome of mortality would bring: it would greatly reduce the number of articles but also allow a more direct comparison of the effects of the proposed interventions.

Widening the concept of outcomes to include a range of clinical, process, and implementation metrics will allow a much richer and more complete assessment of the literature but make comparison much more challenging.

There is also a question about the balance between an inductive as opposed to a hypothetico-deductive approach. Common 'outcomes' may emerge from the literature which challenge those postulated by the authors.

It should be possible to screen out papers which make no mention of any outcome assessment (as proposed page 9 line 4) and then compare the themes which emerge from the literature to those used a priori.

I think some mention also needs to be made that different metrics will

Thank you for the comprehensive set of issues raised about the outcomes. Our primary outcome is mortality. Our secondary outcomes include other clinical outcomes (including major and minor complications), as well as process and implementation outcomes. We have changed the text in the abstract to clarify this: 'Three types of outcomes will be collected including clinical, process and implementation outcomes. The primary outcome will be mortality – all other outcomes will be secondary. Secondary outcomes will include clinical outcomes (major and minor complications), as well as process and implementation outcomes.'

We do think that this is in line with our study objectives as we are looking for studies that improve patient outcomes through improved quality of care and therefore both of these need to be present.

Thank you. This text refers to the description of possible intervention, not the outcome. For example, the intervention may be delivered to staff, but the outcome may be mortality. We have added 'by' in front of 'staff' to clarify this.

This is now consistent with the abstract.

need to be manipulated to allow direct comparison (eg if survival vs mortality is reported, then the sign needs inverting to allow comparison - if data are to be converted into a common format such as 30-day mortality then this should be explained, or if all studies will be excluded apart from those which explicitly use 30 day mortality then this also needs to be stated).

I am certain the authors are very clear on this, but the terminology used throughout the paper could be tightened to make it more explicit to the reader.

Is the primary outcome of mortality a defining characteristic of the search, or a benchmark for comparing the (probably small) subset of papers who report it? If the latter, will this analysis of small subset be the most useful lesson to be gained from this literature?

2. Developing this theme, the proposed 'narrative synthesis' is defensible and reasonably standard as is the use of only descriptive statistics. However an alternative might be something like a 'best-fit framework' analysis which has been used to synthesise qualitative literature (<https://eur03.safelinks.protection.outlook.com/?url=https%3A%2F%2Fbmcmmedicine.biomedcentral.com%2Farticles%2F10.1186%2F1741-7015-9-39&data=01%7C01%7Cnataliya.brima%40kcl.ac.uk%7C5a182748b691430cbdda08d7a0bc50a1%7C8370cf1416f34c16b83c724071654356%7C0&data=Et22SM%2FHoleuc%2BfdcBFdGcJBXvgO57Z%2FKzvFz1nJxgo%3D&reserved=0>). This might allow the authors to assess the extent to which the literature they identify supports their proposed models of clinical, process, or implementation outcomes and, if it does not, critically appraise where their proposed models are challenged. 'Quality' is a difficult term to define and may have components which pull in different directions: it is easy to conceive an intervention that reduces cost, improves access, but worsens individual clinical outcome despite improving population health outcomes. How would the 'quality' of such an intervention be addressed by the current study strategy?

3. This finally leads to the question of bias. The end goal of this research is to ultimately inform the design and delivery of a future intervention (page 13 line 4). As such, learning from failure may be as crucial as learning from success, but there is a huge inherent bias in published accounts of quality improvement efforts whereby unsuccessful attempts, or unwanted side-effects, are rarely published. The greatest bias, to my mind, of this data will be that it will largely synthesise the current experience of success but speak very little to the current experience of failure. I think the authors have taken every care to address bias in their study design, but would prefer to see a greater acknowledgement of these fundamental issues in their discussion of general limitations and also specific issues with bias.

4. The study quite rightly aims to assess both surgical and anaesthetic care, and is also framed very much within the (completely appropriate, in my view) paradigm of health systems strengthening. A key process in systems thinking is the definition of the system of interest and its boundary, and this is not dwelt on in terms of how a health systems strengthening intervention (HSSI) might affect perioperative care. Page 7, line 17-20 give a good idea of the boundaries considered as the 'surgical system' which include anaesthesia, intensive care, and the emergency department. These

Yes, we will be using the GRADE criteria in this review to assess primary clinical outcomes – which is mortality. This has been clarified throughout the paper.

Thank you for these reflections. We agree that a narrow focus on the clinical outcome of mortality would greatly reduce the number of articles and limit the utility of the review.

We recognise that common outcomes may emerge from the literature that do not necessarily align with our clinical, process outcome criteria nor our 8 implementation outcomes. We have added 'other' category to record those outcomes. The 'Data collection sheet' and the text has been added to reflect this: 'Those studies that were collected under the heading 'other', we will look for common themes and then compare these themes to our three defined categories of outcomes.'

We considered this point very carefully. From the pilot screening and during the process of defining our

are appropriately reflected by the search terms in Appendix 1. Once these papers are identified, however, how will the research team navigate this system boundary? Would - for example - an intervention purely aimed at improving ITU care in a hospital which provides surgical care be included, even if no explicit mention of surgery is made? This seems to me to be the hardest part of synthesising this data. I think the answer will be heuristic and difficult to define explicitly before the literature is reviewed. However I think a bit more explicit thought to show how these systems boundaries are intended to be managed would be useful, and the link between deductive vs inductive reasoning developed in line with Comment 1 above. Completely defining all boundaries and terms at the start obviously maximises the ability of the protocol to be repeated, but does not allow for iterative refinement based on exposure to the actual data.

Specific points:

Page 4 line 10: is this the Lancet Commission on health systems? I think a bit more specificity would be useful (eg 'a recent Lancet Commission on high quality health systems').

Page 4 line 49: some of the points raised above need to be somehow addressed in the limitations.

Page 5 line 24: classifies = classified?

Page 5 line 30: less = fewer?

Page 5 line 38-42: although familiar, this is just one model to improve access to surgical care.

Page 6 line 19: these objectives could be revised for clarity in light of major moments above. If you are only assessing the methodological rigour of studies with primary clinical outcomes does this mean you are excluding all studies for which their primary outcome was not clinical? Or just not assessing their rigour? This seems a slightly narrow focus and will limit the ultimate utility of the results.

Page 6 line 56: it might be worth quickly explaining how you will exclude trauma as it seems likely that some 'systems strengthening' efforts may act on multiple surgical services. If an improvement is explicitly focussed on trauma then I am assuming it all be excluded, but what will you do with those which focus on multiple surgical subtypes (such as WHO checklist implementation)? These may include trauma data, so will they be excluded, or will you try to unpick the trauma component of the data, or will you include them as being predominantly 'surgery' focussed as opposed to trauma?

Page 7 line 5: Is this the defining criteria - any study that aims to improve quality AND patient outcomes? To me this is quite different from systems strengthening efforts which might look at staff, family, or institutional outcomes. This is part of the wider question about clarity of outcomes mentioned above.

Page 7 line 17: mention of staff here seems in contradiction to line 5 above?

Page 8 line 57: Are interventional studies the only way to assess system strengthening efforts? This has a very positivist, 'cause-and-effect' construction which is well suited to more common discrete medical interventions (such as pharmaceutical testing). Is this

exclusion and inclusion criteria the studies that do not mention assessment of outcomes are referred to observational studies that mostly audits or descriptive reporting of secondary data collection. In this review we are particularly interested in interventional studies which by definition will have evaluation and assessment of outcomes.

Thank you. The primary outcome is defined as mortality within 30 days. We acknowledge that survival data will require inversion.

We now think the terminology has been tightened.

We agree that the papers with a primary outcome of mortality may represent a small subset and that this is not a defining characteristic of the search, as seen in the search strategy.

Thank you for suggesting the best-fit framework which, as you mention, has been used to synthesise qualitative data. We are not inclined to use this framework as very few hospital quality improvement intervention studies are qualitative studies. We accept that quality can be a difficult concept to evaluate, despite a number of well

necessary if looking to assess quality improvement, system strengthening etc?

established frameworks that are used. Whilst we are not reporting on a quality outcome per se, we will summarise where outcomes (including those that could fit under a descriptor of “quality”) travel in different directions.

Thank you. A bullet point has been added to the Strengths the Limitations section: ‘Although every effort will be made to address the publication bias, we acknowledge that unsuccessful quality improvement efforts and unwanted side-effects, are rarely published. Therefore, this review may not reflect the failed attempts in the field of improvement of quality of surgical care.’

Thank you. We agree that it will be useful to tighten the boundaries at this stage as much as practical and feasible. For the purpose of this systematic review, only studies that mention surgery or surgical patients explicitly will be included. We have added some changes to the text to clarify this: 'For the purpose of this review only studies that explicitly refers to a surgical procedure or a surgical patient will be included.'

This has been changed in the text.

	This has been addressed. This has been changed. This has been changed. Thank you, we agree with this statement. This has been addressed and objectives revised. Thank you. We have amended the text to clarify this. Trauma (both explicitly focusing or included as part of the intervention) will be excluded as it can be considered a separate specialty of surgery and will be covered in a separate systematic review. This has been addressed
--	---

	This has been addressed Thank you, as mentioned above we considered this point very carefully. In this review we are particularly interested in interventional studies.
--	---

VERSION 2 – REVIEW

REVIEWER	Tom Bashford NIHR Global Health Research Group on Neurotrauma, University of Cambridge I know two of the authors personally and teach on the MSc at Kings College London (KCL) which is run by the senior author. I am paid an annual honorarium for this teaching, but have no contractual obligations to either him or KCL. I have no other conflicts of interest.
REVIEW RETURNED	04-Mar-2020
GENERAL COMMENTS	Thank you for the thoughtful and comprehensive responses to my previous points. I have no further changes to suggest to the current manuscript.